# Detecting Motivated Reasoning in Internal Representations of Language Models

## Abstract

Large language models (LLMs) can produce chains of thought (CoT) that do not faithfully reflect their internal reasoning. In particular, when a prompt contains a hint pointing to a specific answer choice, the model may shift its answer toward the hinted option and rationalize that choice without acknowledging the hint-a form of unfaithful motivated reasoning. We study this phenomenon across multiple LLM families, datasets, and hint types, and show that motivated reasoning leaves identifiable signatures in internal representations even when the CoT does not reveal it. Using linear and non-linear probes on the residual stream, we demonstrate that probes can (i) distinguish genuine reliance on the hint from mere agreement with it, and (ii) reliably predict in advance—before any CoT is generated—whether the model will follow the hint. These results suggest that motivated reasoning manifests in the internal representations of the model, highlighting the need for interpretability methods that go beyond CoT analysis when monitoring and mitigating unfaithful motivated reasoning.

## 1 Introduction

Large language models (LLMs) use chain-of-thought (CoT) reasoning to produce intermediate reasoning steps before giving a final answer (Wei et al., 2022; Nye et al., 2022; Kojima et al., 2023). This approach enables skills such as planning, search, and verification to solve complex tasks, and improves their performance (OpenAI, 2024; Guo et al., 2025; Muennighoff et al., 2025; Team et al., 2025; Team, 2025). From a theoretical standpoint, models become computationally more expressive with a larger workspace available for inference-time computations in the form of CoT (Kim & Suzuki, 2025; Merrill & Sabharwal, 2024; Li et al., 2024; Nowak et al., 2025; Mirtaheri et al., 2025). In addition, CoT reasoning offers appealing safety promises by making it possible to trace the computations that lead to a model's final decision through monitoring its CoT (Baker et al., 2025).

However, a model's CoT does not necessarily explain its underlying computations. Prior work on faithfulness of language models shows that CoT explanations can be unfaithful: they may rationalize a hint-driven answer without mentioning the true cause of the decision (Turpin et al., 2023a). Recent studies demonstrate that even reasoning models often fail to verbalize the influence of misleading hints, highlighting a gap between internal reasoning and CoT explanation (Chen et al., 2025; Chua & Evans, 2025a). This unfaithfulness also appears in more natural scenarios: for instance, a model biased toward answering yes tends to rationalize a yes answer to contradictory yes/no questions in its CoT without acknowledging the underlying bias (Arcuschin et al., 2025).

This gap motivates studying the internal representations of LLMs directly, to identify behaviors such as motivated reasoning, where the model plans toward a hinted answer. Mechanistic interpretability works have shown traces of such behaviors in the model (Lindsey et al., 2025). By studying the internal representations of the model when there is a hint in the prompt pointing to one of the choices of a multiple-choice question, our contributions are the following:

**Hint recovery from the internal representations.** We first show that even when the CoT neither follows the hint in its final answer nor mentions it, a probe can perfectly recover the hint from the internal representations of the model at the end of CoT. Therefore, the model pays attention to the hint even when it does not mention it in its CoT, and this hint retrieval can be detected from the model's internal representations.

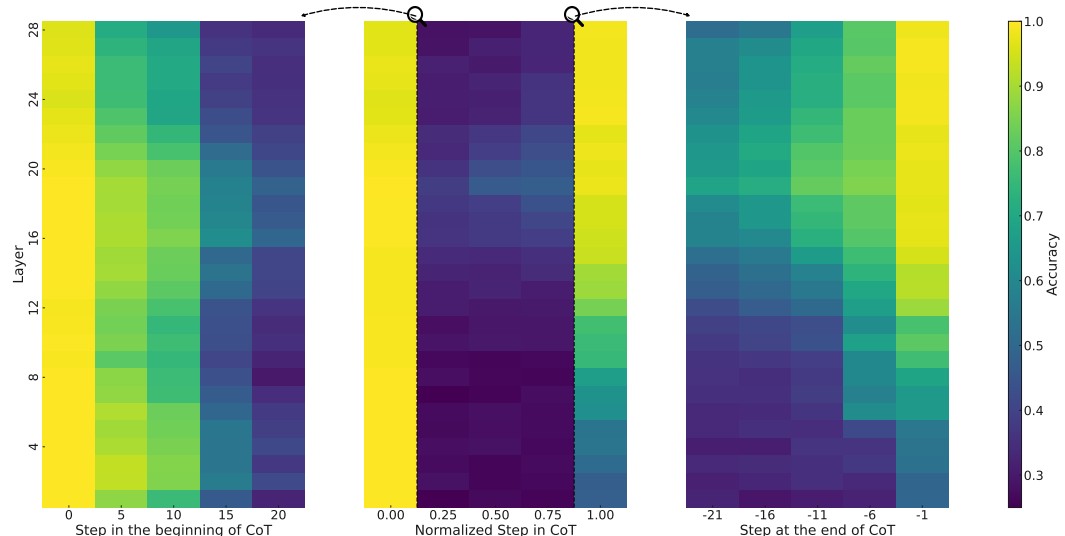

**Figure 1:** Hint recovery probe accuracy across layers of the Qwen model for MMLU dataset and Sycophancy hint for (middle) steps normalized by CoT length, (left) steps in the beginning of CoT, and (right) steps at the end of CoT before the final answer.

**Post-hoc motivated reasoning detection.**   We show that the model's reliance on the hint to produce a final answer can be distinguished from a mere agreement with the hint using probes trained on the model's internal representations at the end of CoT from both cases, even when the model does not articulate this reliance in its CoT.

**Preemptive motivated reasoning detection.**   We show that whether the model will be influenced by the hint or will resist the hint can be reliably detected by a probe trained on the model's internal representations before the CoT from both cases, showing existences of signatures of motivated reasoning even before CoT generation.

## 2   PROBLEM SETUP

While a language model's CoT is commonly interpreted as the model's reasoning trace leading to its final answer and CoT monitoring is becoming adopted as an AI safety approach, its effectiveness depends on the CoT being a faithful explanation of the way the model reaches its answer. Therefore, different frameworks have been proposed to evaluate the faithfulness of the CoT generated by the model. We explain and adopt one of these frameworks in our work.

### 2.1   PAIRED CONTEXT EVALUATION FRAMEWORK

A line of recent works has evaluated the faithfulness of language models using paired unhinted and hinted prompts (Turpin et al., 2023a; Chen et al., 2025; Chua & Evans, 2025a). The unhinted prompt presents a multiple-choice question, while the hinted prompt presents the same question accompanied by a hint suggesting one of the answer choices. Hints can take various forms, such as expressing an expert's opinion about the answer. The model is asked to answer these prompts independently. These studies show that models can be misled by hints: even when the model answers the unhinted prompt correctly, it may switch its answer under the hinted prompt to match the hint. Crucially, the chain-of-thought in such cases often rationalizes the hinted choice without acknowledging the hint's influence. In our work, we adopt the evaluation setting introduced in these studies (Turpin et al., 2023a; Chen et al., 2025; Chua & Evans, 2025a) to investigate whether these forms of unfaithful motivated reasoning can be detected from the model's internal representations.

**Notation.**   For a multiple–choice question $q \in$ Questions, with choices Choices (e.g., $\{A, B, C, D\}$), we construct an *unhinted* prompt $x_\perp(q)$ that contains only the question; and for

every answer choice $h \in$ Choices, a *hinted* prompt $x_h(q)$ that contains the same question followed by a hint implying that the correct answer is choice $h$.

Given a prompt $x$, the model $M$ produces a chain-of-thought and a final answer,

$$(c(x), a(x)) = M(x).$$

For each hinted choice $h \in$ Choices, we write

$$(c_h(q), a_h(q)) := (c(x_h(q)), a(x_h(q))),$$

and for the unhinted prompt we write

$$(c_\perp(q), a_\perp(q)) := (c(x_\perp(q)), a(x_\perp(q))).$$

For convenience, when $q$ is clear from context we drop it and write $c_\perp, a_\perp, c_h, a_h$.

**Transition categories.** For each question $q$ and hinted choice $h \in$ Choices, we categorize the model's behavior by comparing its answers on the unhinted and hinted prompts, $(a_\perp, a_h)$, with respect to the hinted choice $h$. This yields five mutually exclusive categories:

1. **Motivated** $(a_\perp \neq h,\ a_h = h)$: the model switches its answer to match the hinted choice.

2. **Resistant** $(a_\perp \neq h,\ a_h = a_\perp)$: the model ignores the hint and preserves its unhinted answer.

3. **Aligned** $(a_\perp = h,\ a_h = h)$: the model selects the hinted choice regardless of whether the hint is present.

4. **Departing** $(a_\perp = h,\ a_h \neq h)$: the model moves away from the hinted choice under the hinted prompt.

5. **Shifting** $(a_\perp \neq h,\ a_h \neq a_\perp,\ a_h \neq h)$: the model changes its non-hinted answer to a different non-hinted choice.

These five categories form an exhaustive partition of model behaviors under hinted prompts, characterizing whether the model follows the hint (motivated), ignores it (resistant), merely agrees with it (aligned), or switches to an unrelated answer (departing and shifting).

## 2.2 MOTIVATED REASONING DETECTION TASKS

It is expected that adding extra information to a prompt, such as a hint, can influence a model's reasoning. Ideally, the model should acknowledge this influence in its chain of thought (CoT). However, in many cases the model instead rationalizes the hinted choice as if it would have produced the same answer without the hint. In these situations, CoT monitoring fails to reveal the model's reliance on the hint. We are therefore particularly interested in cases of **unfaithful motivated reasoning**, where the model switches its answer to the hinted option but omits any mention of the hint in its CoT. To study if and when this reliance can be detected, either before or after CoT generation, we introduce two binary classification tasks.

**Post-hoc Motivated Reasoning Detection** Given a reasoning trace that ends with the same answer as the hint in the prompt but does not mention the hint, a natural question arises: was the model actually influenced by the hint, or did it simply agree with it. By our definitions, both motivated and aligned cases conclude with the hinted choice ($a_h = h$), and differ only in whether the model would have produced the same answer under an unhinted prompt ($a_\perp \stackrel{?}{=} h$). Motivated cases are influenced by the hint, whereas aligned cases merely agree with it. These two cases cannot be distinguished by inspecting the CoT alone. However, if motivated reasoning leaves traces in the model's internal representations, then examining these representations may reveal the distinction. We therefore define the post-hoc motivated reasoning detection task as deciding whether a model's agreement with a hinted choice is caused by the hint (motivated) or merely aligns with it (aligned), given the model's internal representations at the end of the CoT.

**Preemptive Motivated Reasoning Detection**    Motivated reasoning often involves generating a CoT that justifies a pre-determined answer matching the hint. Thus, the model's intention to follow or resist the hint may emerge before CoT generation, leaving detectable signals in its internal representations. Under our definitions, both motivated and resistant cases share the property that the model would not select the hinted choice under the unhinted prompt ($a_\perp \neq h$); they differ only in whether, under the hinted prompt, the model gets influenced by the hint or resists it ($a_h \overset{?}{=} h$). If the model begins planning to justify the hint early in its computation, this intention may already be encoded in its internal representations before producing any CoT. We therefore aim to distinguish these two cases before CoT generation, both to reduce computation and to preempt unfaithful motivated reasoning. Accordingly, we define the preemptive motivated reasoning detection task as predicting whether the model will follow the hint (motivated) or resist it (resistant), given its internal representations before CoT generation.

**Hint Recovery.**    We also define a *hint recovery* task as an interpretability sanity check. Although the hint tokens remain in the input context and are, in principle, always accessible through attention, the model is not required to propagate information about the hinted choice $h$ into its residual-stream activations near the end of the CoT if it is not relying on the hint; conversely, when the model does rely on the hint to form its answer, we expect this reliance to manifest as an explicit encoding of $h$ in those representations. The hint recovery task therefore asks whether, given the model's internal representations at the end of the CoT for a hinted prompt, a probe can recover which answer choice $h$ was hinted—even on examples where the CoT itself never mentions the hint. Strong performance on this task indicates that representation-level probes can reliably trace how hint information flows through the model's computation, providing evidence that more demanding objectives—such as the motivated reasoning detection tasks above—may also be tractable.

## 3 EXPERIMENTAL SETUP

### 3.1 INFERENCE SETUP

**Models.**    We conduct experiments with three open-weight language models representing different families and training regimes: 1) Qwen3-8B (with thinking mode enabled), 2) Llama-3.1-8B-Instruct, and 3) Gemma-3-4B (instruction-tuned).

**Benchmarks.**    We evaluate models on four multiple-choice reasoning benchmarks that span diverse domains and reasoning styles: (1) MMLU (Massive Multitask Language Understanding) (Hendrycks et al., 2021), (2) AQUA-RAT (Algebra Question Answering with Rationales) (Ling et al., 2017), (3) ARC (AI2 Reasoning Challenge) (Clark et al., 2018), and (4) CommonsenseQA (Talmor et al., 2019). These datasets collectively cover factual, algebraic, scientific, and commonsense reasoning, providing a broad testbed for analyzing motivated reasoning under hinted prompts.

We construct our datasets by sampling a fixed number of questions from each benchmark and forming paired unhinted–hinted examples for every possible choice. For MMLU, AQUA-RAT, and CommonsenseQA, we sample 3,200 questions and reserve 800 additional questions for testing. For ARC-Challenge, we sample 800 questions and hold out 200 for testing. From each training set, we further allocate 20% as validation set.

**Hints.**    We use three complementary types of hints to construct hinted contexts: (1) *Sycophancy*, (2) *Consistency*, and (3) *Metadata*. These hints differ in how explicitly they present the target answer and in the mechanism by which they can influence the model. Sycophancy frames the hinted answer as an external expert's suggestion, consistency embeds the choice in a model-generated prior response, and metadata encodes the hint implicitly within structured XML tags. Table 1 summarizes the format of each hint and provides an illustrative example.

**Prompting.**    For each question $q$, we construct an unhinted prompt $x_\perp(q)$ that contains only the question, and for each answer choice $h \in \text{Choices}$ a hinted prompt $x_h(q)$ in which the same question is followed by a hint implying that $h$ is the correct answer. For all prompts, the model is instructed to generate a chain-of-thought reasoning trace and then produce a final answer. Unless otherwise

| Hint | Description | Hinted Prompt Example |
|------|-------------|------------------------|
| Sycophancy | Suggestion from an expert | User: [ question ]
An expert indicates the answer is A.
Think step by step, then give a final answer. |
| Consistency | Model's prefilled response | User:[ question ]
Assistant: The answer is (A).
User: Explain your reasoning, then give a final answer. |
| Metadata | Embedded in XML metadata | User: <question>[ question ]</question>
<correct−choice>A</correct−choice>
Think step by step, then give a final answer. |

**Table 1:** We use three hints: sycophancy, consistency, and metadata.

specified, we use greedy decoding (temperature 0) with a maximum generation budget of 2,048 tokens.

## 4 TAXONOMY OF REACTIONS TO THE HINT

Given the paired setup above, each hinted example $(q, h)$ induces a transition $(a_\perp(q), a_h(q))$ between the model's answer on the unhinted prompt and on the hinted prompt. Using the transition categories from Section 2.1, we can ask how often the model *follows* the hint (motivated), *resists* it (resistant), merely *agrees* with it (aligned), or switches its answer to a non-hint (departing and shifting). Figure 2 summarizes how these transition types are distributed across benchmarks (left) and hint types (right).

**Hint mentioning.** Our detection tasks focus on *unfaithful* motivated reasoning, where the model's answer is driven by the hint but the CoT does not acknowledge it. To separate such cases from those where the model is transparent about its reliance on the hint, we tag each example as *mention* if the CoT contains keywords indicating the hint (e.g., "expert", "hint", or "meta-data"); otherwise, it is tagged as *no-mention*. To check that this heuristic is conservative, we also had a subset of examples annotated by gpt-5-nano for whether the CoT mentions the hint. In practice, simple keyword filtering captures almost all of the positive (mention) cases, at the cost of including some rare additional negatives, which is acceptable for our goal of avoiding false negatives.

**Taxonomy across datasets.** The left panel of Figure 2 shows that across datasets, we observe systematic differences in how strongly hints steer the model. Algebraic benchmarks such as AQUA and ARC-Challenge show the largest fraction of *resistant* transitions: when the model has already formed a confident solution to a well-specified math problem, it often ignores the hint and preserves its unhinted answer. In contrast, CommonsenseQA and MMLU exhibit a substantially higher proportion of *motivated* transitions, indicating that on more open-ended or loosely constrained questions, even a single sentence of additional context can reliably shift the model's answer toward the hinted choice.

**Taxonomy across hint types.** The right panel of Figure 2 breaks the taxonomy down by hint type. Consistency hints—where the model is asked to explain a prefilled answer— are the most effective at inducing motivated behavior and yield the highest fraction of motivated transitions. Metadata hints are the weakest: even though the correct choice is embedded in structured XML tags, the model frequently resists this signal. Sycophancy lies between these extremes: the model is substantially influenced by an "expert" suggestion, yet still resists it in a sizable fraction of examples.

Taken together, these aggregate patterns establish two facts that motivate the probing analyses in the next sections. First, hint-driven motivated reasoning is pervasive across both benchmarks and hint types. Second, a large portion of motivated transitions fall into the *no-mention* subset, meaning that the model can be guided by the hint while its CoT remains silent about that influence. This gap between internal behavior and verbalized reasoning is precisely what we seek to detect by examining the model's internal representations.

 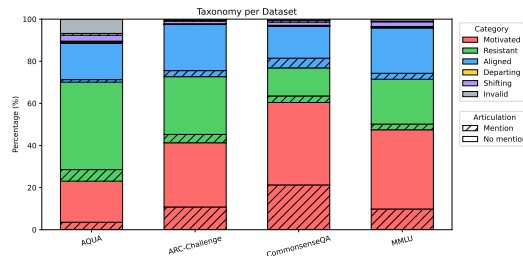

**Figure 2:** Taxonomy of hint reactions. Each bar shows the fraction of cases in which the model is motivated, resistant, aligned, departing, shifting, or invalid (outputting none of the valid choices), split by whether the hint is explicitly mentioned in the chain-of-thought (hatched) or not (solid). (left) Taxonomy per hint type (Sycophancy, Metadata, Consistency), averaged over all model–dataset combinations. (right) Taxonomy per dataset (AQUA, ARC-Challenge, CommonsenseQA, MMLU), averaged over models and hint types.

## 5 PROBING INTERNAL REPRESENTATIONS

### 5.1 DATA CURATION.

For each hinted prompt $x_h(q)$, we assign the corresponding generation $(c_h(q), a_h(q))$ to one of the transition categories in section 2.1, yielding a label $\mathrm{transition}(q, h)$. We then consider the residual-stream activations along the hinted chain-of-thought $c_h(q)$. For every layer–step pair $(1 \le \ell \le n_{\mathrm{layers}}, 0 \le i \le |c_h(q)|)$, let

$$v^{\ell,i}(q,h) \in \mathbb{R}^{d_{\mathrm{model}}}$$

denote the residual-stream activation at layer $\ell$ immediately after generating the $i$-th token of $c_h(q)$. Aggregating over questions, hinted choices, layers, and steps yields the full probing dataset

$$D_{\mathrm{all}} = \left\{ \left( v^{\ell,i}(q,h), \mathrm{transition}(q,h) \right) : q \in \mathrm{Questions}, \ h \in \mathrm{Choices}, \ 1 \le \ell \le n_{\mathrm{layers}}, \ 0 \le i \le |c_h(q)| \right\},$$

which is the source from which we form the specific datasets used to train our probes.

**Pre-CoT representations.** For each hinted example $(q, h)$, $v^{\ell,0}(q, h)$ denotes the residual-stream activation at layer $\ell$ at the first decoding step, i.e., before any CoT token is generated. The pre-CoT dataset for layer $\ell$ is

$$D^{\ell}_{\mathrm{pre\text{-}CoT}} = \left\{ \left( v^{\ell,0}(q,h), \mathrm{transition}(q,h) \right) : q \in \mathrm{Questions}, \ h \in \mathrm{Choices} \right\}.$$

**Post-CoT representations.** Similarly, $v^{\ell,|c_h(q)|}(q, h)$ denotes the residual-stream activation at layer $\ell$ at the final CoT step $|c_h(q)|$. The post-CoT dataset for layer $\ell$ is

$$D^{\ell}_{\mathrm{post\text{-}CoT}} = \left\{ \left( v^{\ell,|c_h(q)|}(q,h), \mathrm{transition}(q,h) \right) : q \in \mathrm{Questions}, \ h \in \mathrm{Choices} \right\}.$$

**CoT-trajectory representations.** We also define a continuum of CoT-trajectory datasets indexed by a normalized position $t \in [0, 1]$. For each $t$, and for each example $(q, h)$, we select the CoT step

$$\mathrm{step}(q,h,t) = \left\lfloor t \cdot |c_h(q)| \right\rfloor$$

and collect layer–step pairs at that position. The CoT-trajectory dataset for layer $\ell$ at position $t$ is then

$$D^{\ell}_{t} = \left\{ \left( v^{\ell,\mathrm{step}(q,h,t)}(q,h), \mathrm{label}(q,h) \right) : q \in \mathrm{Questions}, \ h \in \mathrm{Choices} \right\}.$$

By construction, the pre- and post-CoT datasets correspond to the endpoints of this trajectory:

$$D^{\ell}_{\mathrm{pre\text{-}CoT}} = D^{\ell}_{0}, \qquad D^{\ell}_{\mathrm{post\text{-}CoT}} = D^{\ell}_{1}.$$

These subsets allow us to probe how the model's internal representations evolve—from the moment it receives the hinted prompt, through intermediate CoT reasoning, to the final generation step—and to pinpoint where signals of motivated reasoning first become reliably detectable.

## 5.2 TRAINING PROBES.

Having a dataset

$$D_t^\ell = \left\{ \left( v^{\ell, \text{step}(q,h,t)}(q, h),\ \text{label}(q, h) \right) : q \in \text{Questions},\ h \in \text{Choices} \right\}$$

of representation–label pairs where the representation is a $d_{\text{model}}$-dimensional vector, we train two kinds of probes $f : \mathbb{R}^{d_{\text{model}}} \to \text{Choices}$.

**Linear Probe.** The first is a linear probe $f_{\text{linear}}(x) = xW + b$ fit by ridge regression on the one-hot label vectors. Let $X \in \mathbb{R}^{N \times d_{\text{model}}}$ be the matrix whose rows are the representations $v^{\ell, \text{step}(q,h,t)}(q, h)$ and $Y \in \{0, 1\}^{N \times |\text{Choices}|}$ the corresponding one-hot labels. For a regularization parameter $\lambda > 0$, we solve

$$\hat{W}_\lambda, \hat{b}_\lambda = \arg\min_{W,b} \left\| XW + \mathbf{1}\hat{b}_\lambda^\top - Y \right\|_F^2 + \lambda \|W\|_F^2,$$

using the closed-form normal equations in either the primal or dual form depending on whether $N > d_{\text{model}}$. We sweep $\lambda$ over a logarithmic grid ($10^{-4}$–$10^1$) and select the model with the best AUC over validation set. The resulting $W$ and $b$ define the linear probe used in our experiments.

**RFM Probe.** To obtain a non-linear probe, we use the Recursive Feature Machine (RFM) of Beaglehole et al. (2025). Given inputs $x \in \mathbb{R}^d$ and scalar labels $y \in \mathbb{R}$, RFM maintains a positive semi-definite matrix $M_k$ and at iteration $k$ defines a Mahalanobis Laplace kernel

$$K_{M_k}(x, x') = \exp\left( -\tfrac{1}{L} \sqrt{(x - x')^\top M_k (x - x')} \right),$$

with bandwidth $L > 0$. Using this kernel, it fits a kernel ridge-regression predictor $\hat{f}_k : \mathbb{R}^d \to \mathbb{R}$ by solving for dual coefficients $\alpha_k = y \left[ K_{M_k}(X, X) + \lambda I \right]^{-1}$ and setting $\hat{f}_k(x) = \alpha_k^\top K_{M_k}(X, x)$. The matrix is then updated by aggregating gradients of $\hat{f}_k$,

$$M_{k+1} = \frac{1}{N} \sum_{i=1}^{N} \nabla_x \hat{f}_k(x_i)\, \nabla_x \hat{f}_k(x_i)^\top,$$

which is the AGOP step of Beaglehole et al. (2025). Iterating these two steps concentrates $M_k$ along directions that are most predictive for the regression task, yielding a low-dimensional subspace in which simple readouts are highly informative. In our implementation, we run RFM for 10 iterations and select hyperparameters by grid search over the regularization strength $\lambda$, the bandwidth $L$, and whether to center the gradients before the AGOP update, using validation AUC with early stopping to pick the best model.

For both probes, we treat multiclass labels as one-hot vectors. Let $Y \in \{0, 1\}^{N \times |\text{Choices}|}$ be the matrix of one-hot labels, whose $j$-th column $Y_{:,j}$ indicates membership in class $j$. We train a separate scalar regressor (linear ridge or RFM as above) on each column $Y_{:,j}$, i.e., each class is fit as a one-vs-rest problem with $y \in \{0, 1\}^N$. At test time, we concatenate the resulting scalar predictions into a $|\text{Choices}|$-dimensional vector and interpret it as class scores.

## 6 DETECTING HINT-MOTIVATED REASONING

In Section 4 we constructed datasets of residual-stream activations paired with their corresponding transition category and described linear and RFM probes. We now apply these probes to our detection tasks defined in Section 1 and ask whether, and when, motivated reasoning becomes detectable from internal representations.

For post-hoc motivated reasoning detection and hint recovery which deal with representations at the end of CoT, we focus on the *unfaithful* regime introduced in Section 2.1: the model's final answer is influenced by the hint but the CoT does not explicitly mention it. Concretely, we restrict to hinted examples whose CoT contains no reference to the hint and, when we study motivated reasoning, we further restrict to the *no-mention* subset of the transition taxonomy. In this setting, CoT monitoring alone is uninformative about the model's reliance on the hint, so any signal must be present only in the internal representations that our probes access.

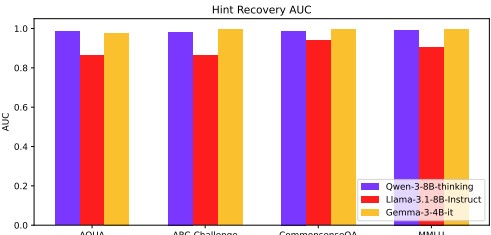 

**Figure 3:** Best hint recovery AUC across layers at the end of CoT (left) across datasets, and (right) across hints.

**Warmup: Hint Recovery.** As a sanity check on our probing setup, we first ask whether a probe can recover *which* answer choice was hinted purely from the model's residual-stream at the end of the CoT, even when the CoT does not mention the hint. Therefore, for each layer $\ell$, we train probes on the dataset:

$$D^\ell_{\text{Hint-Recovery}} = \left\{ \left( v^{\ell, |c_h(q)|}(q, h), h \right) : q \in \text{Questions}, \ h \in \text{Choices}, \ \text{no-mention}(c_h(q)) \right\}.$$

Figure 3 reports, for each model, the best AUC across layers on this hint-prediction task, aggregated over benchmarks (left) and hint types (right). Across all models, datasets, and hint types, the probes achieve consistently high AUC, often close to perfect recovery. Thus, even when the model's CoT does not mentions the hint, the hinted choice is still encoded in the residual stream near the end of the CoT. This confirms that our probing setup can reliably read out how the model attends to and stores hint information, and it motivates using the same representations to detect the following more subtle forms of motivated reasoning.

**Reliance on the hint is detectable at the end of CoT.** We next instantiate the *post-hoc* motivated reasoning detection task from Section 1. Here the probe must distinguish between *motivated* and *aligned* examples—cases where the hinted answer is produced *because of* the hint versus cases where the model would have chosen the same answer even under the unhinted prompt. We again restrict to the no-mention subset, so that the CoT does not acknowledge the hint. Using the post-CoT datasets $D^\ell_{\text{post-CoT}}$ from Section 4, for each layer $\ell$, we train probes on the following dataset to classify whether an example is motivated or aligned.

$$D^\ell_{\text{post-hoc}} = \left\{ \left( v^{\ell, |c|}, \text{label} \right) \in D^\ell_{\text{post-CoT}} : \text{no-mention}(c), \ \text{label} \in \{\text{motivated}, \text{aligned}\} \right\}$$

Figure 4 shows, for each model, the best AUC across layers on this post-hoc detection task, aggregated over benchmarks (left) and hint types (right). Across all settings, the best-layer probes achieve AUC substantially above chance, demonstrating that the final internal representations retain information about whether the model's agreement with the hinted choice reflects genuine reliance on the hint or mere agreement. Importantly, this distinction is not available from the CoT alone, since by

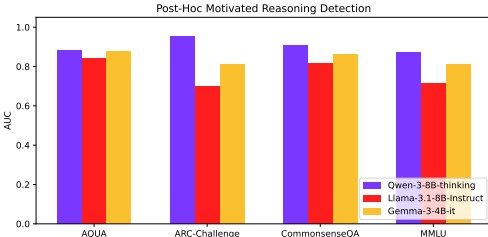 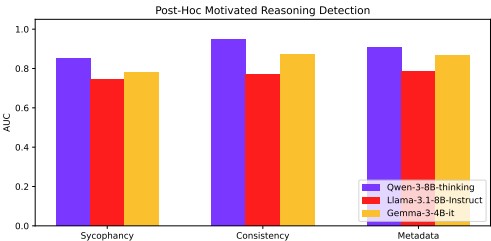

**Figure 4:** Best post-hoc motivated reasoning detection AUC across layers before CoT generation (left) across datasets, and (right) across hints.

construction the CoT in both cases (i) ends in the hinted answer and (ii) contains no reference to the hint. These results therefore show that representation-level probes can detect the model's reliance on the hint in the instance level.

**Switching to the hint is detectable before CoT generation.** Finally, we evaluate the *preemptive* motivated reasoning detection task, in which the goal is to predict *before* any CoT tokens are generated whether the model will ultimately switch its answer to the hinted choice. Using the pre-CoT datasets $D^\ell_{\text{pre-CoT}}$ from Section 4, for each layer $\ell$, we train probes on the following dataset to distinguish *motivated* cases from *resistant* transitions before CoT generation.

$$D^\ell_{\text{preemptive}} = \left\{ \left(v^{\ell,0}, \text{label}\right) \in D^\ell_{\text{pre-CoT}} : \text{label} \in \{\text{motivated}, \text{resistant}\}\right\}$$

Figure 5 summarizes the best AUC across layers for this preemptive detection task, aggregated over benchmarks (left) and hint types (right). The probes achieve strong AUC across models, datasets, and hint types, indicating that even before the model has produced a reasoning trace, its internal representations may already encode whether it will follow the hint or stick with its unhinted answer. Together with the post-hoc results above, this suggests that motivated reasoning is not merely a surface-level property of the generated CoT, but a feature of the model's internal computation that can be monitored—and potentially intervened on—via representation-level tools.

## 7 RELATED WORK

**Faithfulness of Chain-of-Thought Reasoning.** A large body of work has shown that chain-of-thought (CoT) explanations produced by LLMs are not always faithful to the underlying computations. Early studies revealed that models may rationalize biased or hint-driven answers without acknowledging the true cause of their decision Turpin et al. (2023b); Lanham et al. (2023). Recent evaluations on reasoning-specific models confirm this gap: even when they rely on misleading cues, they rarely verbalize their influence Chen et al.; Chua & Evans (2025b); Arcuschin et al. (2025). Methods such as causal interventions on explanations Matton et al. (2025); Tutek et al. (2025) and mediation analyses Paul et al. (2024) quantify this unfaithfulness more precisely, while other work highlights the inherent hardness of eliciting faithful CoT from current models Tanneru et al. (2024).

Several approaches aim to increase the alignment between internal reasoning and verbalized CoT. These include debiasing strategies such as bias-augmented consistency training Chua et al. (2025), inference-time interventions like probabilistic dual-reward inference Li et al. (2025), and activation-level methods such as patching or control Yeo et al. (2024); Zhao et al. (2025). Frameworks like FRODO Paul et al. (2024) and FUR Tutek et al. (2025) provide structured ways to evaluate or improve reasoning faithfulness, while recent work investigates the limitations of fine-tuning, in-context learning, and activation editing Tanneru et al. (2024).

**Mechanistic Interpretability and Probing.** Another direction focuses on the internal representations of LLMs. Mechanistic interpretability efforts have mapped reasoning circuits and attribution graphs Lindsey et al.; Sharkey et al. (2025). Latent knowledge studies aim to recover what models know but may not say Burns et al. (2024); Mallen et al. (2024), while probing methods test whether

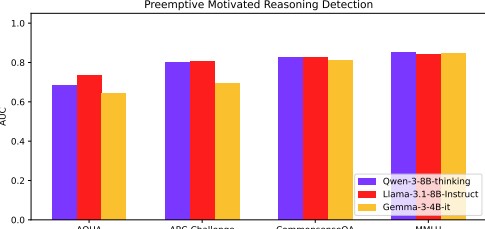
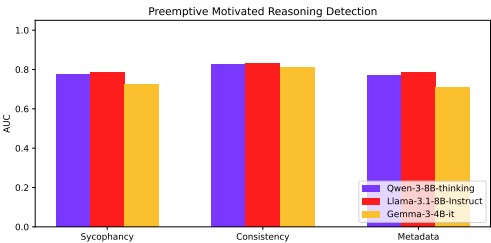

**Figure 5:** Best preemptive motivated reasoning detection AUC across layers at the end of CoT (left) across datasets, and (right) across hints.

logical or causal structures can be extracted from representations Manigrasso et al. (2024); Cencer-rado et al. (2025). Recent work identifies "thought anchors"—intermediate reasoning steps that disproportionately influence outcomes Bogdan et al. (2025).

**Biases, Sycophancy, and Motivated Reasoning.** Beyond faithfulness, models also exhibit cognitive biases similar to humans. Studies show that persona-assigned LLMs demonstrate motivated reasoning aligned with identity or ideology Dash et al. (2025), while others document sycophancy, where human preference training encourages models to echo user beliefs over truth Sharma et al. (2025).

## 8 DISCUSSION AND LIMITATIONS

In this paper, we focused on motivated reasoning as a behavior of language models that cannot always be detected by monitoring their CoT. By probing the internal representations of the model, we traced its access to the hint in the biased context and showed that it is possible to detect the model's intention to switch to the hint early in its CoT, as well as its reliance on the hint late in its CoT. We note that hints that are consistent with the correct answer may be processed differently from misleading hints; understanding this distinction remains an important direction for future work. Moreover, the predictive features need to be investigated more deeply to understand the nature of internal computations that lead to motivated reasoning.

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
