# OpenReview forum: "Detecting Motivated Reasoning in Internal Representations of Language Models"
_ICLR.cc/2026/Conference — Submitted to ICLR 2026_

### Official Review · Reviewer_GuYp · 2025-10-27

**Soundness:** 3
**Presentation:** 3
**Contribution:** 3
**Rating:** 6
**Confidence:** 3

**Summary:**

This paper uses probes to detect unfaithful reasoning in LLMs. They find that probes are able to detect unfaithful reasoning, including before it happens, using model internals.

**Strengths:**

* The motivation of this paper is great - unfaithful COT reasoning is a major problem with one of the most promising approaches for AI safety, COT monitoring. This approach, if sound, could help to flag when normal COT monitors which look at the model’s text output are potentially unreliable. I also think the motivation of being able to detect unfaithful reasoning without having to run counterfactuals is strong
* This paper uses a strong experimental set up, building off of one of the standard ways of evaluating and demostrating unfaithful COT.

**Weaknesses:**

* How the probe is trained is pretty unclear to me from the description, and it feels like the results could be widely varying in interesting-ness depending on how it was trained
* Less a weakness and more an area for further impact: I think there’s a huge opportunity to design a probe for unfaithful reasoning in general. If there’s any way to generalize this to that use case, I think this work could be hugely impactful. Then, looking at the results of what gets flagged would be quite interesting
* This isn’t a huge deal, but the paper is only 6 pages, so it could be worth extending the results or doing additional analysis, since there’s certainly a lot more to be done here.
* It’s not clear to me how much of the work is being done by having something model internals based (a probe) vs. just having a good dataset for training a classifier (which could just be a finetune of the whole model or some other model). It’d be helpful to know if the internals specifically are adding value over these other baselines, to know how much value the internals are adding. I’d guess the internals-based approach isn’t necessary (but I think the results are interesting if done by a probe or a full-param finetune)

**Questions:**

1. Can you expand in more detail on how the data for the probe was produced?
2. Can the method in the paper be extended to detecting unfaithful reasoning in general? If so, how would that be done?

---

### Official Review · Reviewer_Q9Jw · 2025-10-29

**Soundness:** 2
**Presentation:** 1
**Contribution:** 2
**Rating:** 2
**Confidence:** 4

**Summary:**

This paper focuses on the problem of faithful reasoning of large language models (LLMs). Specifically, it examines the internal representation of LLMs to analyze the influence of bias on the model's reasoning in its chain-of-thought. The empirical results show that bias remains consistently predictable/recoverable from representation at the end of CoT, and probes can help distinguish reliance on the bias reliably in advance.

**Strengths:**

1. The focused problem of the unfaithfulness of LLMs' reasoning and bias detection is important and of high significance.
2. The findings empirically show that the probe is a useful tool to detect the bias reliance in model internal representation.

**Weaknesses:**

1. The presentation is not good and clear enough that allow readers to easily catch the challenge and technical contribution of this work.
2. The bias term is not well-defined and there is limit intuitive example or illustration on what kind of context would be catergorized into bias, and the literature search is not sufficient for surveying previous related work on bias detection in nlp context.
3. The probe technique is not presented clearly to let the reader understand the detailed implementation.
4. The Figures are confusing and hard to read without specific experimental description on specific findings. Please consider to enrich the caption to add more intuitive explanation.
5.  Is there any baseline methods with CoT to show the basic detection performance?

Not about the main content:
1. As the authors claimed, they use LLM to implement some methods that they can verified, I'm not sure whether it should be categorised into LLM-engaged research and allowed by the submission.

**Questions:**

1. Please consider enhancing the representation to explicitly state the technical challenge of the existing research problem.
2. Please do a more thorough literature survey on the bias context detection problem in both the conventional NLP domain and LLM research area.
3. Please consider involving more models and other baselines to make the experiments convincing.

---

### Official Review · Reviewer_v32V · 2025-10-30

**Soundness:** 2
**Presentation:** 3
**Contribution:** 2
**Rating:** 6
**Confidence:** 2

**Summary:**

The paper investigates potential misalignment between Chain-of-Thought (CoT) and "internal computations" of LLMs in biased contexts. Using "RFM (Recursive Feature Machine) probes" on the "residual stream" of Qwen3-8B, Llama-3.1-8B, and Gemma-3-4B, the authors study three tasks: (i) "Bias recovery"—at the end of CoT, the hint is decodable from internal representations even when the output/CoT does not mention or follow it; (ii) "Retrospective detection"—distinguishing "motivated" from "coincident" cases that look identical from CoT text alone; and (iii) "Prospective detection"—predicting whether the model will follow the hint "before" generating CoT. The setting covers three hint types (sycophancy/consistency/metadata) and four datasets (MMLU, AQUA-RAT, ARC-Challenge, CommonsenseQA), with a four-way transition taxonomy (resistant/motivated/coincident/divergent).

**Strengths:**

1. Clear Motivation: CoT-only monitoring may not reveal **motivated reasoning**; probing internal states is positioned as a complementary signal for safety/evaluation.
2. Broad Experimental Coverage: Multiple models/datasets and three hint types suggest that the studied effects occur across settings; taxonomy distributions are provided.
3. Methodological Simplicity : External probes on the residual stream without LLM retraining; training protocol (per-layer/step probes; 80/20 split) is straightforward.

**Weaknesses:**

1. What do the probes capture? (interpretability/construct validity) RFM has relatively high capacity. Strong “warm-up” hint prediction near the CoT end could be influenced by **lexical/positional traces** of the hint rather than an internal representation of “intention.” Causal interventions (e.g., activation patching, position shuffling, explicit hint masking) would help test whether suppressing hint traces substantially reduces retrospective/prospective performance.
2. Probes appear trained per layer/step. **Cross-dataset/hint/model** transfer (e.g., train on MMLU and test on ARC-Challenge; train on sycophancy and test on metadata; train on one model and test on another) is not reported. Such results would indicate whether the method captures more general signals rather than distribution-specific patterns.
3. Since CoT-only cannot distinguish **motivated** vs. **coincident** by construction, a strong **CoT-only** classifier baseline (e.g., style/length features) and/or comparisons with recent faithfulness frameworks (e.g., FRODO (Paul et al., 2024); FUR (Tutek et al., 2025)) would contextualize the added value of internal-representation probing.
4. Figures emphasize “best layer” trends, but multi-seed means ± std, confidence intervals, or significance tests are not shown. Adding error bars for the main AUC/accuracy plots (e.g., Fig. 1–2 and the detection figures) would strengthen the statistical basis.
5. Providing RFM hyper-parameters, regularization, optimizer/schedule, epochs/early-stopping, random seeds, and split scripts (including whether splits are stratified) would facilitate independent reproduction.
6. “Motivated reasoning” can be broader than sycophancy/consistency/metadata biases. Clarifying how the four-way taxonomy connects to this term may aid interpretation.

**Questions:**

1. How do "linear probes" or shallow MLPs perform relative to RFM?
2. Have you tried "input masking/position shuffling" to assess whether probes primarily read "hint-token residue"?
3. Do you have "cross-dataset/hint/model" transfer results (e.g., train on one dataset/hint/model, test on another)?
4. At which exact point is the "prospective" representation extracted (post-question, pre-first CoT token)? A "layer × step" heat map would be informative.
5. How does the method compare to "CoT-only" baselines and to faithfulness frameworks such as "FRODO" (Paul et al., 2024) and "FUR" (Tutek et al., 2025), both cited in the related work?

---

### Official Review · Reviewer_G4NV · 2025-11-01

**Soundness:** 2
**Presentation:** 2
**Contribution:** 2
**Rating:** 4
**Confidence:** 3

**Summary:**

This paper investigates the phenomenon of motivated reasoning in LLMs, focusing on cases where biased contexts influence the model’s reasoning and answers. The authors applied probes on internal activations and conducted study over different benchmarks.

**Strengths:**

This paper studies the faithfulness of LLMs' CoT reasoning, which is an important topic to discuss. It examines over different models and benchmarks, and provided lucid illustrations.

**Weaknesses:**

1. It is unclear how this work differs from previous works on LLM faithfulness, and there are already varieties of work demonstrating LLM's unfaithfulness from the perspective of internal activations including but not limited to [1][2][3]. Could you clarify your novelty upon these works?
2. The generality of the probe across different task domains is unclear.
3. The paper did not provide deeper analysis or explanation as why the bias happens. Is it related to the model architecture or training methodology?
4. The paper doesn't explicitly discuss how the detection methods can be applied as mitigation strategies.
5. More datasets and models should be incorporated for evaluation.
6. Interpretability analysis is limited. Empirical experiments are a bit rudimentary. At least some case studies should be provided.

The paper's main text is only of six pages. This itself is not a weakness but it's clear that much more experiments should be included before publication at top-tier conferences like ICLR. The current version should be submitted as a workshop paper or short paper instead.

[1] Discovering Latent Knowledge in Language Models Without Supervision (ICLR 2023)

[2] The Internal State of an LLM Knows When It’s Lying (EMNLP 2023 Findings)

[3] Overthinking the Truth: Understanding how Language Models Process False Demonstrations (ICLR 2024)

**Questions:**

Please see weaknesses.

---

### Author Response · Authors · 2025-12-04
**General Response**

We thank all reviewers for the careful and constructive feedback. We are glad that reviewers found the problem of unfaithful / motivated CoT reasoning important and the experimental setup (multiple models, datasets, hint types) promising. In the revised manuscript, we have substantially clarified our problem formulation, expanded the technical details of the method, and improved the presentation and examples. We summarize the main clarifications and changes below.

**Presentation**

Clarifying “hint”, “bias”, and “motivated reasoning”.
We agree that our earlier terminology around “bias” was confusing. In the revision we now:

Use “hint” consistently to denote the extra prompt component that suggests an answer choice (sycophancy, consistency, metadata). Table 1 gives concrete formats and examples for each hint type.

Define “motivated reasoning” in a narrow, operational sense: cases where the model switches from a non-hinted answer under the unhinted prompt to the hinted answer under the hinted prompt (motivated), and where the CoT does not mention the hint (the “no-mention” subset).

Introduce “aligned” (previously “coincident”) and explicitly separate “aligned” vs “motivated” vs “resistant” vs “departing/shifting” categories, so that the reader can see exactly which regimes our probes are asked to distinguish.

We also now explain more clearly that we focus on the “no-mention” regime where CoT monitoring is, by construction, uninformative; we identify these examples using keyword filters and a sanity check with an LLM annotator.

In response to concerns about clarity, we have:

Rewritten Sections 2–3 to introduce the paired-context framework, taxonomy, and detection tasks more gradually, with explicit notation for questions, hints, and transitions.

Clarified the prose around Figures 1–5 and expanded their captions so that the reader can understand the main takeaways without cross-referencing multiple sections.

Added more detailed descriptions of the hints and representative prompt templates (Table 1), and emphasized that our focus is on “no-mention” motivated reasoning where CoT monitoring fails.

We acknowledge that more qualitative case studies would be valuable; we have already generated several representative examples and will include a subset in the appendix (space permitting) or in a companion repository.

**What do the probes capture? (interpretability / construct validity).**

To address concerns about what signal the probes actually use, we added a hint recovery task as a warm-up and sanity check: even when the CoT never mentions the hint and the final answer does not follow it, a probe can still recover the hinted choice from the residual stream at the end of the CoT with very high AUC across models, datasets, and hint types (Fig. 3). This confirms that our setup can reliably read out how hint information flows into internal representations.

We then show that on top of this, the same representations support:

Post-hoc detection: distinguishing motivated vs aligned examples at the end of the CoT, even though both end with the hinted answer and both CoTs omit the hint (Fig. 4).

Preemptive detection: distinguishing motivated vs resistant examples before CoT generation, from the residual stream at the first decoding step (Fig. 5).

We fully agree that this does not yet prove that the probes have isolated a clean “intention to follow the hint” subspace; they may still rely on subtler lexical or positional traces of the hint. We now say this explicitly in the discussion and frame our results as evidence that some behaviorally relevant signal about hint-following is present in internal representations, not as a complete causal explanation. In future work, we plan to complement these results with activation patching, location masking, and position shuffling ablations to better distinguish genuine intention-like features from residual lexical features.

**Baselines and CoT-only methods.**

Reviewers rightly asked about CoT-only baselines and comparisons to frameworks like FRODO/FUR. In our core detection tasks, a strong CoT-only classifier is degenerate by construction:

In the post-hoc task, motivated and aligned examples are matched on the hinted prompt, final answer, and CoT text (we filter to “no-mention” cases), so any classifier that only sees the CoT cannot do better than chance.

In the preemptive task, the representation is taken before any CoT token has been generated.


**Probe architecture and training details; linear vs RFM.**
We substantially expanded Section 5.2 to give a more concrete description of how probes are trained:

We describe the linear probe explicitly as ridge regression with one-vs-rest readouts, and specify the λ grid and selection by validation AUC.

We give algorithmic details for RFM (kernel choice, AGOP-style updates, number of iterations, grid over λ, bandwidth L, centering, and early stopping).

---

> ### Author Response · Authors · 2025-12-04
>
> We have revised the text to state this explicitly and to explain why we do not report a CoT-only baseline on those tasks: CoT features are intentionally factored out so that we can isolate what is added by internal representations. At the same time, we agree that CoT-only methods and frameworks like FRODO and FUR are highly relevant for broader faithfulness, and we now emphasize the complementarity: our probes are meant as a low-overhead monitoring signal that could be combined with these frameworks in future work (e.g., using probe scores to flag cases where explanation-level methods may be unreliable).
>
> We also make the data pipeline more reproducible by specifying dataset sizes, train/validation/test splits, and where representations are taken (per-layer, at pre-CoT / trajectory positions / end-of-CoT). We agree that adding error bars or confidence intervals would further strengthen the statistical claims; if the paper is accepted, we will include multi-seed error bars in the camera-ready version.
>
> Presentation and examples.
> In response to concerns about clarity, we have:
>
> Rewritten Sections 2–3 to introduce the paired-context framework, taxonomy, and detection tasks more gradually, with explicit notation for questions, hints, and transitions.
>
> Clarified the prose around Figures 1–5 and expanded their captions so that the reader can understand the main takeaways without cross-referencing multiple sections.
>
> Added more detailed descriptions of the hints and representative prompt templates (Table 1), and emphasized that our focus is on “no-mention” motivated reasoning where CoT monitoring fails.
>
> We acknowledge that more qualitative case studies would be valuable; we have already generated several representative examples and will include a subset in the appendix (space permitting) or in a companion repository.
>
> LLM usage.
> One reviewer raised a question about whether this should count as “LLM-engaged” research. For transparency: we used LLMs only for editing (minor rewriting of paragraphs) and for small scripting assistance (e.g., generating plotting boilerplate). All experimental design, datasets, probe training code, and analyses were implemented, run, and verified by the authors. We do not rely on an external LLM for any of the core scientific claims.
>
> We hope these clarifications address the main concerns about novelty, construct validity, baselines, generality, and presentation. We believe the revised manuscript more clearly positions our contribution as showing that signatures of hint-motivated reasoning are encoded in internal representations before and after CoT generation, in a setting where CoT text alone is intentionally uninformative, and that this opens a concrete path toward representation-level monitoring and, in future works, mitigation.

---

### Meta-Review · Area_Chair_z4YJ · 2026-01-10

**Summary:**

The key concerns raised by reviewers can be grouped as:
* Novelty and usefulness: related to prior work and how the detection methods an be used as mitigation strategies
* Validity of the construction: generality across tasks; what do the probes capture;
* Understanding of results: Causal analysis missing; needs for additional experiments and ablations; deeper insights required
* Presentation: This has been a recurring concern

**Reviewer Concerns:**

* Novelty: The rebuttal mostly reframes the intent. That helps the conceptual framing but doesn’t fully answer novelty relative to existing work.
* Validity of the construction: The authors added a hint recovery task as a sanity check, however the concern is deeper; it helps that the authors acknowledge this, however since this is a key concern related to the core of the method, more extensive ablation studies would be needed.
* Understanding of results: multi-seed error bars and confidence intervals are not included as requested. A stronger causal analysis would also be helpful to increase convincingness.
* Presentation. Multiple clarifications and extensive rewriting have improved this. At the same time, it is fair to acknowledge that a paper going from 6 pages to 9+ pages constitutes major re-writing, and it might be preferable to re-assess it as a fresh submission.

**Reviewer Scores:**

Given the above, I would expect only reviewer Q9Jw to increase their scores by 1 point, given the improvements in presentation, with the caveat that -as discussed earlier- such a major rewriting might be better re-assessed as a new submission in the future.

---

### Decision · Program_Chairs · 2026-01-26

Reject